# *Vibrio splendidus* AJ01 Promotes Pathogenicity via L-Glutamic Acid

**DOI:** 10.3390/microorganisms11092333

**Published:** 2023-09-17

**Authors:** Ya Li, Weibo Shi, Weiwei Zhang

**Affiliations:** 1Collaborative Innovation Center for Zhejiang Marine High-Efficiency and Healthy Aquaculture, Ningbo University, Ningbo 315832, China; 2School of Marine Sciences, Ningbo University, Ningbo 315832, China

**Keywords:** *Vibrio splendidus*, L-Glu, transcriptomic analysis, virulence

## Abstract

*Vibrio splendidus* is a pathogen that infects a wide range of hosts, especially the sea cucumber species *Apostichopus japonicus*. Previous studies showed that the level of L-glutamic acid (L-Glu) significantly increased under heat stress, and it was found to be one of the best carbon sources used by *V. splendidus* AJ01. In this study, the effects of exogenous L-Glu on the coelomocyte viability, tissue status, and individual mortality of sea cucumbers were analyzed. The results showed that 10 mM of L-Glu decreased coelomocyte viability and increased individual mortality, with tissue rupture and pyknosis, while 0.1 mM of L-Glu slightly affected the survival of sea cucumbers without obvious damage at the cellular and tissue levels. Transcriptomic analysis showed that exogenous L-Glu upregulated 343 and downregulated 206 genes. Gene Ontology (GO) analysis showed that differentially expressed genes (DEGs) were mainly enriched in signaling and membrane formation, while a Kyoto Encyclopedia of Genes and Genomes (KEGG) analysis showed that DEGs were significantly enriched in the upregulated endocytosis and downregulated lysosomal pathways. The coelomocyte viability further decreased by 20% in the simultaneous presence of exogenous L-Glu and *V. splendidus* AJ01 compared with that in the presence of *V. splendidus* AJ01 infection alone. Consequently, a higher sea cucumber mortality was also observed in the presence of exogenous L-Glu challenged by *V. splendidus* AJ01. Real-time reverse transcriptase PCR showed that L-Glu specifically upregulated the expression of the *fliC* gene coding the subunit protein of the flagellar filament, promoting the swimming motility activity of *V. splendidus*. Our results indicate that L-Glu should be kept in a state of equilibrium, and excess L-Glu at the host–pathogen interface prompts the virulence of *V. splendidus* via the increase of bacterial motility.

## 1. Introduction

The Splendidus clade is the largest in the *Vibrionaceae* family. A recent study confirmed that the Splendidus clade consists of 19 species, the members of which are often associated with the mortality of marine animals, causing huge economic losses [1]. *Vibrio splendidus,* a member of the Splendidus clade, is distributed throughout the world and is the main Vibrio species in marine environments [1,2]. In previous outbreaks of various aquatic animal epidemics, *V. splendidus* has been proven to be an important strain in different mortality events, resulting in great losses to the worldwide aquaculture industry [3]. *V. splendidus* is an opportunistic pathogen that infects fish such as turbot (*Scophthalmus maximus*) [4]; echinoderms such as the sea cucumber sp. *Apostichopus japonicus*; and marine bivalves such as the scallop sp. *Patinopecten yessoensis* [5], the pacific oyster sp. *Crassostrea gigas* [6], the clam sp. *Ruditapes decussatus* [7], and the mussel sp. *Mytilus* [8].

It has been reported that skin ulcer syndrome (SUS) in sea cucumbers infected by *V. splendidus* has caused over 80% mortality and over 30% economic losses in China and Japan [3]. *V. splendidus* mainly affects immune-signaling-related pathways in *A. japonicus*, such as endocytosis, lysosomes, the mitogen-activated protein kinase (MAPK) signaling pathway, and chemokines [9]. Many virulence factors of *V. splendidus* have been identified, including the invasion porin OmpU [10], metalloprotease Vsm [11], invasive vesicular serine protease [12], collagenase [13], and type-six secretion system (T6SS)-related proteins [14]. Flagellar C (FliC) is the primary subunit of the flagellum and has been demonstrated to be linked to the virulence of *V. splendidus* AJ01, mediating not only the adhesion to sea cucumber coelomocytes [15] but also the host immune responses [16]. In addition to these virulence factors, several amino acids (AAs) have been proven to mediate the virulence of pathogens that infect sensitive hosts in pathogen–host interaction models [17]. AAs are involved in different biochemical pathways and play metabolic and physiological roles [18]. Not only does the host rely on AA metabolism to achieve a defense response against a pathogen, but the pathogen also uses AA metabolism to its advantage [18]. Therefore, some AAs are central points of competition between the host and pathogen [19,20]. Pathogens require AAs to fulfill their physiological functions, and the availability of AAs significantly impacts the growth of pathogens as well as the expression of their virulence factors [21]. It has been generally acknowledged that L-glutamic acid (L-Glu) plays an important role in nutrient metabolism, energy supply, immune response, oxidative stress, and signal regulation in host cells and their pathogens [22]. For example, L-Glu can be used as an exogenously added nutrient source to promote the growth of *Pseudomonas syringae* pv. tomato DC3000 [23]. The plant pathogen *Ralstonia solanacearum* utilizes plant-derived L-Glu to promote the production of virulence factors, thus increasing the virulence [24].

It has been previously reported that higher temperatures elevate the levels of threonine, alanine, arginine, glutamic acid, tyrosine, histidine, and glycine [25], among which L-Glu is one of the best carbon sources for the growth of *V. splendidus* AJ01 [26]. These previous results lead us to wonder whether increased L-Glu levels could elevate the virulence of *V. splendidus* AJ01. In the present study, the effects of L-Glu on the cell viability and gene expression profile of coelomocytes in sea cucumbers were determined. The effects of exogenous L-Glu on the virulence and expression of virulence-related genes were determined. Our results show that L-Glu mediates the interaction between the host and pathogen and influences the outcome of infection.

## 2. Materials and Methods

### 2.1. Animals and Bacterial Strain

The sea cucumbers were commercially farmed animals obtained from the Dalian Pacific Aquaculture Company (Dalian, China). The animal experiments were conducted following the recommendations outlined in the National Institutes of Health’s Guidelines for the Care and Use of Laboratory Animals. The experimental protocol was approved by the Laboratory Animal Ethics Committee of Ningbo University (ethical approval number 11436).

*V. splendidus* AJ01 is an opportunistic pathogen that was isolated from *A. japonicus* with skin ulcer syndrome in our previous study. It was deposited as vs. in the China General Microbiological Culture Collection (CGMCC, Beijing, China) (strain no. 7.242).

### 2.2. Coelomocyte Viability Assay

The primary sea cucumber coelomocytes were cultivated referring to the previously used laboratory methods [27]. Briefly, the coelomic fluid was centrifuged at 800× *g* for 5 min, and the harvested coelomocytes were resuspended in Leibovitz’s L-15 medium (Invitrogen, Waltham, MA, USA) supplemented with 0.39 M NaCl. One hundred microliters of cell suspension was added to 96-well culture microplates and cultured at 16 °C overnight. Different concentrations of L-Glu were added into the wells, while Leibovitz’s L-15 medium was added as a control. To determine the effects of *V. splendidus* AJ01 on the viability of coelomocytes in the presence of L-Glu, *V. splendidus* AJ01 at a concentration of 1 × 10^5^ CFU/mL was added to the coelomocytes and incubated overnight with or without L-Glu for 4 h. Ten microlitres of CCK-8 reagent (APExBIO, Houston, TX, USA) was added to wells to be tested and incubated at 16 °C for 2 h, and the absorbance at 450 nm was measured. The absorbance of *V. splendidus* AJ01 incubated in Leibovitz’s L-15 medium was subtracted to exclude the absorbance of *V. splendidus* AJ01 itself. The viability of the coelomocytes was calculated according to the manufacturer’s instructions.

### 2.3. Histological Analysis

To explore the effects of different concentrations of L-Glu on the body wall status, sea cucumbers incubated in seawater containing different concentrations of L-Glu for 72 h were taken for histological analysis [28]. Sea cucumbers without any treatment were used as a control. The tissue samples of each sample were fixed in 10% neutral formaldehyde fixative for 24 h, rinsed in 70% alcohol, and then dehydrated according to concentration gradients of 70%, 85%, 90%, 95%, and 100%, respectively. The tissue samples were clarified in xylene, embedded in paraffin with an average melting temperature of 56 °C, and sectioned at a thickness of 7 µm with a microtome (KD-3358). The tissue sections were deparaffinized and stained with hematoxylin for 10–20 min and eosin for 3–5 min. The sections dehydrated with serial dilutions of ethanol were clarified in xylene and sealed with neutral glycerol. The sections were then observed under a Axio Vert microscope A1 (Zeiss, Oberkochen, Germany).

### 2.4. Immersion Infection Experiment

The method used for the infection experiment referred to Li et al.’s description [29]. To analyze the effect of L-Glu on the survival of sea cucumbers, L-Glu was added to seawater at concentrations of 0.1 mM, 1 mM, and 10 mM, respectively. The status of the sea cucumbers was observed daily, and the number of deaths was recorded. To analyze the effect of L-Glu on *V. splendidus* AJ01 infection, sea cucumbers were divided into three groups, where one group was treated with 1 × 10^7^ CFU/mL *V. splendidus* AJ01 cells in seawater without L-Glu and another group was treated with the same volume of *V. splendidus* AJ01 cells in seawater containing 0.1 mM L-Glu, whereas the group incubated with only 0.1 mM L-Glu was used as the control group.

### 2.5. Transcriptomic Library Construction

The sea cucumbers were evenly divided into two groups, where one group was placed in seawater containing 0.1 mM L-Glu, while another group was used as a control. After soaking for 72 h, the coelomic fluid was collected, and coelomocytes were harvested via centrifugation at 800× *g* for 5 min. The total RNA was isolated according to the manufacturer’s instructions (Vazyme, Nanjing, China). Library construction and high-throughput sequencing were performed by Novogene Biotech (Beijing, China). The clean reads were compared with the *A. japonicus* genome database ASM275485v1 using hisat2 [30]. Gene counts were normalized using DESeq2 software (version 1.20.0) [31]. We screened for differentially expressed genes (DEGs) to detect a > 2-fold variation in the standards. Gene Ontology (GO) and Kyoto Encyclopedia of Genes and Genomes (KEGG) were analyzed using the Cluster Profile software (version 3.8.1) with significant enrichment at a padj < 0.05 threshold.

### 2.6. Real-Time Quantitative Reverse Transcription PCR (qRT‒PCR)

To analyze whether the expressions of virulence factors were regulated by L-Glu, the mRNA levels of five virulence factors identified in *V. splendidus* AJ01, namely, *vspC* [13], *vsm* [32], *vshppd* [33], *hop* [34], and *fliC* [15], were detected using qRT-PCR. The primers were designed based on the nucleotide sequences of the open reading frame of each gene listed in the Appendix A and are listed in Table 1. The amplification efficiencies of the primers were confirmed to be practicable, ranged within 90–110%, according to the method in the MIQE (Minimum Information for Publication of Quantitative Real-Time PCR Experiments) guidelines. To collect the samples, *V. splendidus* AJ01 was cultured to an OD_600_ of approximately 0.5, washed twice using sterilized phosphate-buffered saline (PBS) with a 3% salinity, and divided into four aliquots. The first aliquot was used as a control, the second aliquot was supplemented with 10 mM L-Glu, the third aliquot was supplemented with 100 μL of coelomic fluid from sea cucumbers, and the fourth aliquot was simultaneously supplemented with 10 mM L-Glu and 100 μL of coelomic fluid. Then, all samples were incubated at 28 °C for 1 h, and the cell pellets were collected after centrifugation.

The extraction and reverse transcription of total RNA from bacterial cells was carried out according to the manufacturer’s instructions (Vazyme, China). qRT‒PCR was performed with an ABI7500 instrument (Applied Biosystems, Waltham, MA, USA). Each reaction contained 10 μL of 2 × ChamQ SYBR qPCR Master Mix (Vazyme, China), 0.4 μL of 10 μM each forward and reverse primer, and cDNA transcribed from 1 μg of RNA. The qRT-PCR (with a 20 μL reaction volume) was performed as follows: 95 °C for 30 s followed by 40 cycles at 95 °C for 10 s, 60 °C for 30 s, followed by melting curve analysis at 95 °C for 15 s, 60 °C for 60 s, and then 95 °C for 15 s. The melting curve was used to distinguish whether a sample had non-specific amplification. Relative fold mRNA expression levels were determined using the 2^(−ΔΔCt)^ method [35]. Gene expression was normalized using 16S rDNA as an internal reference gene.

### 2.7. Growth Measurement

*V. splendidus* AJ01 was grown in a 2216E medium at 28 °C in filtered seawater containing 5 g/L of tryptone, 1 g/L of yeast extract, and 0.01 g/L of FePO_4_. To analyze the effect of L-Glu on the growth of *V. splendidus* AJ01, different concentrations of L-Glu were added to an M9 minimal medium with 3% salinity as the sole carbon source [26], and then the absorbance of the culture was recorded at 600 nm. To determine the growth of *V. splendidus* AJ01 in the coelomic fluid containing L-Glu, the filtrated and cell-free coelomic fluid at a ratio of 1% or 10% (*v*/*v*) was added into an M9 minimal medium with 3% salinity containing 10 mM L-Glu, respectively. The absorbance at 600 nm was measured with a Microplate Reader (FlexA-200, Allsheng, Hangzhou, China) under the above incubation conditions. Each growth was repeated in triplicate.

### 2.8. Swimming Motility Analysis

The method for measuring bacterial swimming motility has been previously described [29]. *V. splendidus* AJ01 was cultured in a 2216E liquid medium to the logarithmic growth phase (OD_600_ ≈ 0.5). After washing with PBS with 3% salinity, the swimming motility was detected via needle inoculation on a soft M9 minimal medium with 3% salinity containing 0.3% agar with 10 mM L-Glu as the sole carbon source. To analyze the effect of the coelomic fluid on the bacterial swimming motility of *V. splendidus* AJ01, the bacteria were inoculated into a 0.3% agar medium with 10% filtered coelomic fluid from the sea cucumbers. Each group was repeated in triplicate.

### 2.9. Data Accession Number

The raw data of the transcriptomic sequence has been deposited in the NCBI Short Read Archive (SRA) with accession number SRP446997.

## 3. Results

### 3.1. The Effect of L-Glu on Sea Cucumbers

The effect of L-Glu on sea cucumbers was analyzed at both the cellular and individual levels. When 10 mM L-Glu was added to coelomocytes, the cell viability decreased significantly (Figure 1A). At the individual level, the survival percentage gradually decreased as the concentration of L-Glu increased (Figure 1B). Furthermore, the histopathological observation of the body wall sections showed that L-Glu exhibited different degrees of tissue damage with tissue rupture and pyknosis in a dose-dependent manner (Figure 2). Combining the results of the above experiments, it can be concluded that 10 mM L-Glu showed negative effects on coelomocytes and tissues and at the individual level, resulting in reduced cell viability, tissue damage, and an extremely rapid decrease in survival. The damage to the host gradually became inconspicuous as the concentration of exogenous L-Glu decreased. These results indicate that L-Glu had a dose-dependent effect on sea cucumbers and that excess L-Glu was detrimental to host survival.

### 3.2. L-Glu Affected the Immune-Related Pathways in Sea Cucumbers

In the coelomocytes collected from individuals treated with L-Glu, 343 genes were significantly upregulated, and 206 genes were significantly downregulated (Figure 3A). Three parts of the GO database enriched the hierarchy, namely, cellular components (CCs), molecular functions (MFs), and biological processes (BPs). In the BP part, the term with the most enriched DEGs was regulation of cell communication, and the most significantly enriched terms were ADP-ribosylation factors (ARFs), protein signal transduction, and Ras protein signal transduction. In the CC part, the extracellular matrix (ECM) term was enriched for the most enriched DEGs and was the most significantly different CC. In the MF part, scavenger receptor activity, cargo receptor activity, and enzyme binding were the top three entries (Figure 3B). Two pathways were significantly enriched in the KEGG analysis. Nine upregulated DEGs were significantly enriched in the endocytosis pathway, while seven downregulated GEGs were significantly enriched in the lysosome pathway (Figure 3C,D).

### 3.3. L-Glu Promoted the Virulence of V. splendidus AJ01

The viability of coelomocytes and the bacterial infection experiments were analyzed to determine the role of L-Glu in pathogen–host interactions. *V. splendidus* AJ01 significantly reduced the viability of cultured primary coelomocytes and led to a decrease in the survival percentage of sea cucumbers. The cell viability and survival percentage were further reduced in the presence of L-Glu (Figure 4). The viability of the coelomocytes reduced to 79% after adding *V. splendidus* AJ01, and the viability further decreased to 20% with the addition of L-Glu. Therefore, L-Glu significantly increased the virulence of *V. splendidus* AJ01 in coelomocytes. These results suggest that L-Glu contributed to the virulence of *V. splendidus* AJ01 in its host sea cucumber.

### 3.4. L-Glu Promoted the Growth and Swimming Motility of V. splendidus AJ01

To show whether L-Glu could promote the propagation of *V. splendidus*, bacterial growth was first determined in the presence of L-Glu. *V. splendidus* AJ01 did not show obvious growth when L-Glu was added at a low concentration of 1 mM. As the concentration of L-Glu increased, *V. splendidus* AJ01 showed obvious growth. However, the growth of *V. splendidus* AJ01 was completely inhibited when the L-Glu concentration was increased to 50 mM (Figure 5A). *V. splendidus* AJ01 could not grow in media supplemented with 1% (*v*/*v*) or 10% (*v*/*v*) of coelomic fluid. However, when *V. splendidus* AJ01 was inoculated into the M9 minimal medium with 3% salinity containing different volumes of coelomic fluid and L-Glu, the growth of *V. splendidus* AJ01 increased with the amount of coelomic fluid, which differed from the growth in the coelomic fluid or L-Glu (Figure 5B).

The virulence factors were mainly upregulated in the presence of coelomic fluid, except that the expression of the *fliC* gene was downregulated; however, among the tested genes, only the mRNA level of the *fliC* gene was significantly upregulated in the presence of L-Glu. The expression of the *fliC* gene was also upregulated in the presence of L-Glu in the coelonic fluid matrix, when compared to its expression in the absence of L-Glu (Figure 6A). In addition, the motility of *V. splendidus* AJ01 was promoted in the simultaneous presence of coelomic fluid and L-Glu as the volume of coelomic fluid increased (Figure 6B,C). The growth and motility of *V. splendidus* AJ01 under the above conditions indicate that L-Glu promoted the response of *V. splendidus* AJ01 to coelomic fluid by increasing its proliferation ability and bacterial cell motility.

## 4. Discussion

L-Glu is a multi-functional AA that plays a key role in amino acid metabolism [22]. In studies on the effects of L-Glu on aquaculture animals, it is mainly used as a feed additive and has been found to improve the growth performance and development of animals [36]. For example, dietary supplementation with 2% Amino Gut (a food-grade mixture of Glu and Gln) improved the survival rate of juvenile Nile tilapia [37], while another study focused on its effects on immunity [38]. Our present study first proposed the inhibitory effect of exogenously redundant L-Glu on the cell viability of coelomocytes from sea cucumbers, which was confirmed by the results of tissue damage and reduced survival rates. These results are similar to those of L-Glu-treated A1 astrocytes, where excess L-Glu induces neuronal hyperexcitability and excitotoxicity [39]. In this study, the DEGs of sea cucumbers immersed in seawater containing L-Glu were analyzed to determine the direct effect of L-Glu on the immunity of sea cucumbers.

Endocytosis is the process of translocating extracellular substances into the cell [40], and *V. splendidus* has been shown to enter the coelomocytes via endocytosis [41]. Therefore, the upregulation of the endocytosis pathway suggested that more *V. splendidus* was internalized into the coelomocytes of sea cucumbers. Meanwhile, the lysosomal pathway mainly uses a variety of proteases to digest a variety of macromolecules [42], through which *V. splendidus* can be recognized by the host and cleared through the lysosomal pathway [43]. Thus, the downregulation of lysosomal-related proteases leads to impaired lysosomal function and increases the burden on the host. The upregulation of the endocytosis pathway combined with the downregulation of the lysosomal pathway in sea cucumbers with exogenous L-Glu suggested an attenuated immune-related ability to lyse bacteria during infection, thus promoting bacterial invasion. In our transcriptomic analysis, AFR-related proteins were significantly upregulated and significantly enriched in the endocytosis pathway. It has been shown that ARF-related proteins can affect signaling pathways and participate in the regulation of endocytosis [44]. In addition, studies have shown that ARF proteins mediate susceptibility to a variety of pathogens [45], which is consistent with our hypothesis that L-Glu may promote the infection of *V. splendidus* AJ01 in sea cucumbers.

Both the host and pathogen can influence AA availability to their respective advantages [21]. Certain amino acids, such as arginine, influence the competition between hosts and pathogens because pathogens compete with the host for these amino acids through certain strategies [46]. There have been some sporadic reports of L-Glu-mediated interactions between hosts and pathogens. L-Glu upregulated the expressions of pathogen-associated molecular patterns in the host, while it increased the colonization and the expressions of virulence-related genes of extracellular polysaccharide production, cellulase activity, swimming ability, and biofilm formation in *R. solanacearum* [47]. In our study, L-Glu also promoted *V. splendidus* AJ01 infection; however, contrary to the upregulated cholera toxin production in *Vibrio cholerae* [48], L-Glu only upregulated the expression of the *fliC* gene in *V. splendidus* to increase the bacterial swimming ability in the host environment, which has been well defined as a virulence factor mediating pathogenicity [16].

## 5. Conclusions

In the present study, the effects of exogenous L-Glu on aspects of both the host and pathogen were analyzed. L-Glu affected the host immunity in a dose-dependent manner, leading to the upregulation of the endocytosis pathway and downregulation of the lysosomal pathway at higher additive concentrations. Furthermore, L-Glu was not only used by *V. splendidus* AJ01 as a carbon source but also increased its bacterial swimming motility, which promoted the infection of sea cucumbers by *V. splendidus*. The results of this study demonstrate that L-Glu affected the infection outcome by regulating the immune responses of the host as well as the growth and motility of the pathogen.

## Figures and Tables

**Figure 1 microorganisms-11-02333-f001:**
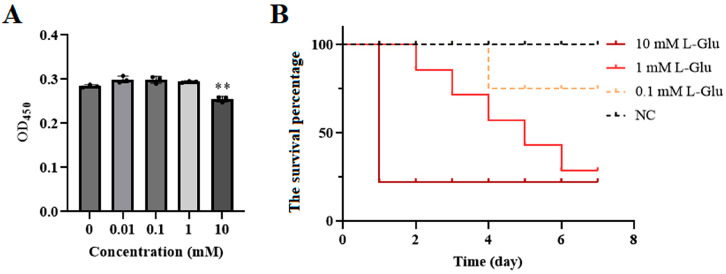
Effects of different concentrations of L-Glu on sea cucumbers at both the cellular and individual levels. (**A**) Viability of coelomocytes treated with different concentrations of L-Glu for 12 h. Data are means and standard deviations from three independent experiments. ** *p* < 0.01. (**B**) The survival percentages of sea cucumbers treated with different concentrations of L-Glu. The dead sea cucumbers in each group were recorded and removed to calculate the survival percentages.

**Figure 2 microorganisms-11-02333-f002:**
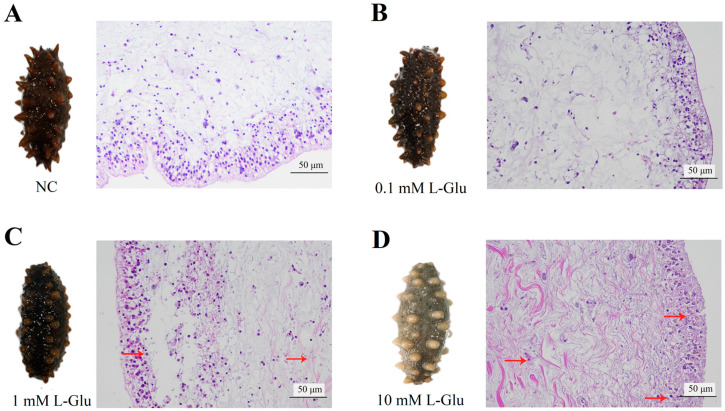
Tissue structure of the body wall of the L-Glu-treated individuals. (**A**) Control sea cucumbers (NC). (**B**–**D**) represent the slices of individual morphological changes and body wall tissues of sea cucumbers treated with different concentrations of L-Glu. The assays were repeated in triplicate, and one representative figure is shown for each sample. Red arrows indicate obvious tissue damage of pyknosis and loosening of tissue connections.

**Figure 3 microorganisms-11-02333-f003:**
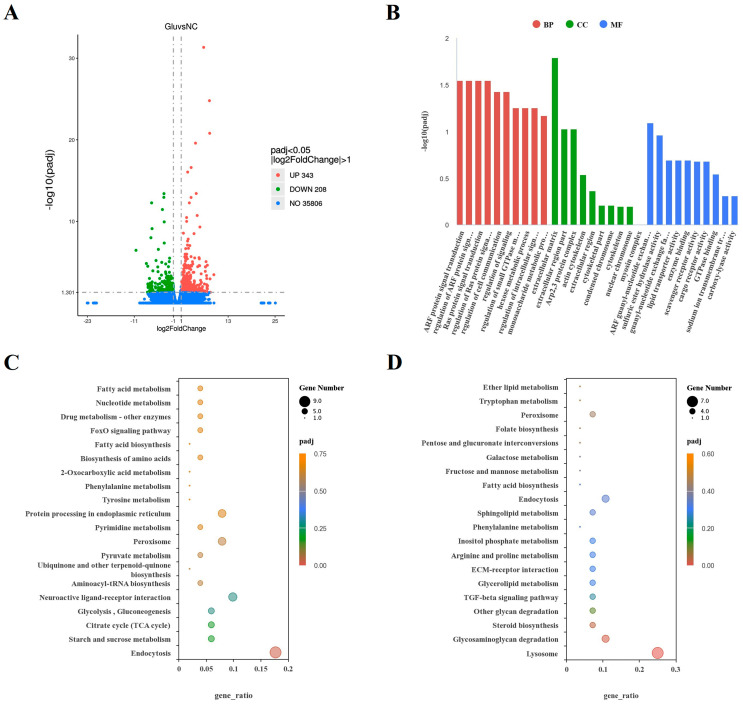
Transcriptomic analysis of sea cucumbers treated with 0.1 mM L-Glu. (**A**) The volcano map of DEGs. The abscissa coordinate is the log2FoldChange value, and the ordinate is −log10padj. The dashed gray line represents the threshold line of DEGs’ screening criteria. (**B**) Histogram based on the GO analysis. The abscissa is the GO element, and the ordinate is the significance level of GO element enrichment, represented by −log10(padj). (**C**) Scatter plot based on KEGG analysis of upregulated DEGs. (**D**) Scatter plot based on KEGG analysis of downregulated DEGs. The abscissa is the ratio of the number of DEGs annotated in the KEGG pathway to the total number, and the ordinate is the KEGG pathway.

**Figure 4 microorganisms-11-02333-f004:**
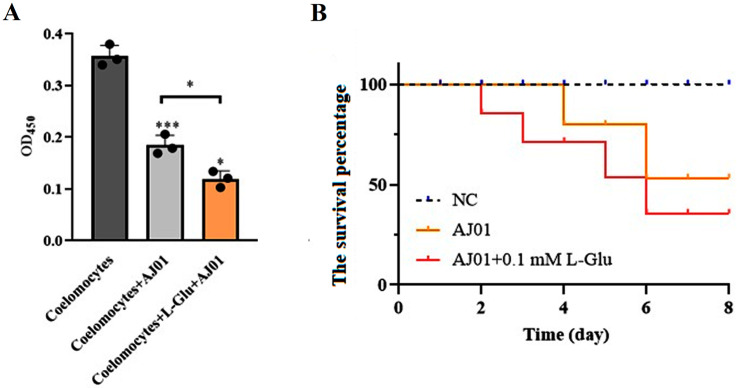
The effects of L-Glu on cell viability and individual survival. (**A**) Cell viability in the presence of *V. splendidus* AJ01 and the *V. splendidus* AJ01 plus L-Glu. Data are means and standard deviations from three independent experiments. * *p* < 0.05 and *** *p* < 0.001. (**B**) The survival percentage of sea cucumbers in the presence of L-Glu when sea cucumbers were treated with *V. splendidus* AJ01. Each growth was performed in triplicate. Data are means and standard deviations from three independent experiments.

**Figure 5 microorganisms-11-02333-f005:**
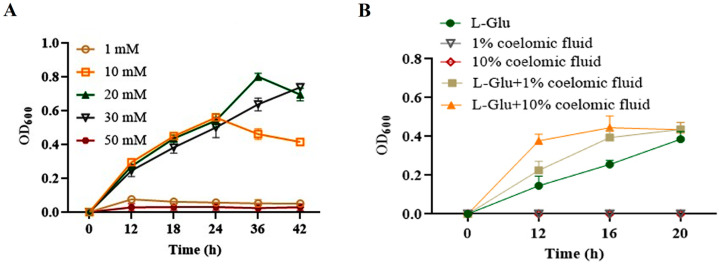
(**A**) Effects of different concentrations of L-Glu on growth of *V. splendidus* AJ01*. (***B**) Growth curve of *V. splendidus* AJ01 in the M9 minimal medium with 3% salinity simultaneously with L-Glu and coelomic fluid. Each growth was performed in triplicate. Data are means and standard deviations from three independent experiments.

**Figure 6 microorganisms-11-02333-f006:**
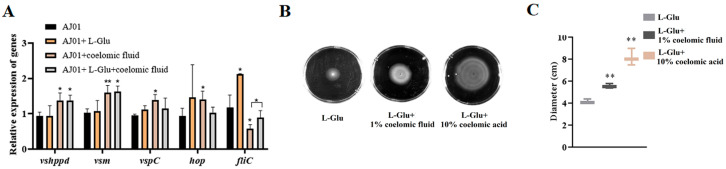
The effects of L-Glu on the expression of virulence-related factors in the matrix of coelomic fluid. (**A**) Expression of virulence-related genes in the presence of L-Glu and coelomic fluid. (**B**) *V. splendidus* AJ01 swimming in the M9 minimal medium with 3% salinity simultaneously with L-Glu and coelomic fluid. (**C**) Statistical analysis of the diameters of the swimming circles. Data are means and standard deviations from three independent experiments. * *p* < 0.05 and ** *p* < 0.01.

**Table 1 microorganisms-11-02333-t001:** Primers used in this study.

Primer	Sequence (5′→3′)
933F	GCACAAGCGGTGGAGCATGTGG
16SRTR1	CGTGTGTAGCCCTGGTCGTA
qtvspCF	GACAGAAACACCGACACCTCC
qtvspCR	CATTCTCCGCATTGTCACTCT
qtvsmF	AAACGAAAGTCCGCTACCA
qtvsmR	CCATTGACCCGAACACCT
qtfliCF	TACCGACTACGCCAAAGAAA
qtfliCR	CCCAGTAAGGTTAAGGCAAGA
qthopF	GAGGCGAACTATGACTTTTCTGAG
qthopR	TCTTCAGCCCATACAATCCA
qtvshppdF	GCCAAGCACCGTTCAAAAGA
qtvshppdR	CGAATGTTTTGATGGTCGGTAT

## Data Availability

The datasets generated and/or analyzed during the current study are available from one of the corresponding authors, Weiwei Zhang, upon reasonable request.

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
