# Peer review of "Vibrio splendidus AJ01 Promotes Pathogenicity via L-Glutamic Acid"

_microorganisms, 2023, doi:10.3390/microorganisms11092333_

Round 1
Reviewer 1 Report
Manuscript ID: microorganisms-2560768
Title: Vibrio splendidus AJ01 promotes pathogenicity by L-glutamic acid
The manuscript investigated the in vitro effects of exogenous L-Glu on virulence of V. splendidus, tissue status and individual mortality of sea cucumbers. With a minor revision this manuscript can be accepted for publication in the journal Microorganisms.
- In the Abstract: write the full names of GO and KEGG analyses.
- In subsection 2.1. Animals, add the ethical approval number if available.
- Title of the subsection 3.1., it should be “The effect of L-Glu on sea cucumber”.
- Mention the origin of V. splendidus AJ01.
- Fig. 1A, the OD reading should be 600, not 450???. The same Fig. 4A.
- The references within the texts are absent. Revise it well following the journal format.
almost fine. Just minor revision is required.
Reviewer 2 Report
The research concerns the effects of Vibrio gorgeousus, the main pathogen that infects a wide range of hosts, in particular the sea cucumber Apostichopus japonicus and analyzes the effects of exogenous L-glutamic acid (L-Glu) on coelomocytes viability, tissue status and mortality individual sea cucumber. The Authors demonstrated that should be maintained in a state of equilibrium and that excess L-Glu at the host-pathogen interface would induce virulence of V. Splendida by increasing bacterial motility.
The subject is interesting and the manuscript well structured. There are problems with citations, but if the authors can address these issues, then publication might be considered. All cited references appear to be relevant for the research, but since they are not visible in the manuscript text where the words "Error! Reference source not found" appear, it is not possible to evaluate the correct citation.
Author Response
The research concerns the effects of Vibrio splendidus, the main pathogen that infects a wide range of hosts, in particular the sea cucumber Apostichopus japonicus and analyzes the effects of exogenous L-glutamic acid (L-Glu) on coelomocytes viability, tissue status and mortality individual sea cucumber. The authors demonstrated that should be maintained in a state of equilibrium and that excess L-Glu at the host-pathogen interface would induce virulence of V. splendidus by increasing bacterial motility.
The subject is interesting and the manuscript well structured. There are problems with citations, but if the authors can address these issues, then publication might be considered. All cited references appear to be relevant for the research, but since they are not visible in the manuscript text where the words "Error! Reference source not found" appear, it is not possible to evaluate the correct citation.
Response: Thanks for your time to review the manuscript. We apologize for the messed format of citations in the original manuscript, which have been carefully corrected according to the journal format.
Reviewer 3 Report
The authors present a study investigating the influence of L-glutamate on the sea cucumber/ V. splendidus AJ1 infection system. While an interesting premise, there are several methodological issues to consider.
The authors present many different concentrations of V. splendidus in this work – 10^5 CFU/ml was added to coelomocytes, 10^7 CFU/ml was added to tank water for cucumber infection, and OD600 = 0.5 was used for gene expression analysis. Why are so many different concentrations/levels of V. splendidus used? Provide logic for these choices so they do not appear to be random. Along these lines, state how V. splendidus was grown for each test because this information is often missing.
The section on RT-PCR lacks much of the information critical for reproducibility and does not come close to conforming to the MIQE guidelines. There are no numbers related to reaction size, cDNA amount, primer concentration, cycle parameters, or how the result was validated (melt curve vs. gel). There are no statements or curves of amplification efficiency of primers or benchmarking of any kind. There is no logic behind using the 16S as a reference gene and no validation for this choice – there are entire papers published on defining genes for RT-PCR normalization in Vibrio (e.g., https://doi.org/10.1371/journal.pone.0144362). Picking a single reference gene and calling it acceptable for normalization does not constitute validation (and citing previous publications that use similarly flawed logic are also insufficient). Particularly, picking a reference gene that is known to exist in multicopy in Vibrio genomes (with nucleotide differences between the multiple copies, which influences amplification efficiency) is not best practice for comparison with single-copy genes.
It remains a question as to how the authors designed primers for their RT-PCR gene analysis when a genome of V. splendidus AJ1 does not appear to be available and their recent Frontiers in Microbiology (https://doi.org/10.3389/fmicb.2023.1127018) paper mapped reads to the Vibrio atlanticus LGP32 genome. This either means that their strain is misclassified or the actual nature of the fragments they are amplifying remains in question. Were the RT-PCR products sequenced and compared to some reference based on high nucleotide similarity? If so, which genome was used because these virulence factors may differ between strains (and V. splendidus and V. atlanticus are at least 5% ANI different even if they’re in the same clade)? These decisions do not appear to be backed by logic and are not explained in the text.
In some sections, the preparation of V. splendidus AJ1 would greatly influence the data. Based on the description of V. splendidus in Bergy’s Manual, this species requires at least 1% NaCl for growth. Due to this requirement, our experience with marine Vibrio strains, and documented changes from the literature, incubating Vibrios in lower NaCl concentrations results in changes in gene expression, changes in viability, and inactivation. In the RT-PCR section, V. splendidus AJ1 was incubated in PBS for an hour before harvest and RNA extraction – PBS has a low NaCl concentration, which will change gene expression in response to a hypotonic solution. That means that the RT-PCR results collected are influenced by the solution and not solely by the addition of glutamate, which is not an adequately controlled experiment. Similarly, the swimming motility and growth of V. splendidus AJ1 on M9 medium will not be possible because of the very low NaCl concentration in M9 medium. We usually use artificial seawater or Instant Ocean to make a minimal medium for Vibrios to ensure that it does not stress the cells with improper solute concentrations. There is no way that V. splendidus AJ1 would grow in M9 medium due to the inadequate NaCl level (see the species description in Bergy’s) unless, of course, V. splendidus AJ1 is misidentified.
Fix your citations – your citation manager messed up.
Line 33- update your citation of the Splendidus clade with the most recent paper (Vibrio Clade 3.0, 2022, Current Microbiology)
The medium is Leibovitz's L-15 medium (misspelled as “Leiboviz's”)
The letters in the figures are tiny and need to be bigger so the reader can easily see them.
The paper is not well written and the introduction is especially complicated. This work requires significant revision for grammar and appropriate word use.
Author Response
Please see the attachement.
